# Clear Yet Crossed: Athletes’ Retrospective Reports of Coach Violence

**DOI:** 10.3390/bs14060486

**Published:** 2024-06-08

**Authors:** Sima Zach, Shlomit Guy, Rinat Ben-Yechezkel, Liza Grosman-Rimon

**Affiliations:** 1Levinsky-Wingate Academic College, Wingate Institute, Tel Aviv 6937808, Israel; rinrin1973@gmail.com (R.B.-Y.); l.grosman.rimon@gmail.com (L.G.-R.); 2School of Physical Education, The Kaye College of Education, Beersheba 8414201, Israel; guyshlomit1@googlemail.com

**Keywords:** coach-athlete violence, psychological violence, physical violence, verbal violence, starvation and food fattening, non-proportional punishing

## Abstract

Aims: This study aimed at examining coach-athlete violence based on the retrospective reports of adults who had been athletes as children and adolescents; predict variables that contribute to the existence of such violence; describe difficulties that the participants encountered as young athletes following such violence; and reveal the outcomes of such violence on their emotions and behaviors, in the past and present. Methodology: The applied mixed methods comprised quantitative self-reporting questionnaires and qualitative interviews. The former included 440 participants (mean age 27.6) who had trained for at least one year in a sports union youth department; the latter included 14 participants (aged 23–37). These competitive athletes came from eight branches of sports. The interviews were analyzed based on the Narrative Approach. Results: According to the quantitative study, all participants had experienced coach violence at least once during their career, mainly psychological violence and neglect, followed by physical violence. Sexual violence was least reported. The age of their retirement from sports and the number of coaches that they had had were significant predictors of violence. Thematic analysis of the qualitative interviews resulted in six types of coach-athlete violence: (1) psychological violence; (2) verbal violence; (3) starvation and food fattening; (4) non-proportional punishing; (5) physical violence; and (6) sexual violence. Conclusions: It is vital that coach-athlete violence is addressed in public discourse, that the topic of young athletes’ safety is introduced into coaching education, and that a position holder is nominated to be in charge of such safety in all sports organizations.

## 1. Introduction

Physical activity is known to have a beneficial impact on people’s physical and emotional health and wellbeing [1,2]. However, it can also have negative outcomes, such as physical injuries, especially in athletes who train at high levels and frequencies [3]. An additional yet much less researched negative impact of physical training relates to athletes who are subjected to maltreatment and even violence [4,5]. Such verbal or physical abuse may include a range of aggressive behaviors directed towards athletes by their peers, parents, coaches, and other professional staff members [6]. Moreover, interpersonal violence against young athletes has been associated with mental health problems and reduced quality of life in adulthood [7].

Not surprisingly, this phenomenon is often kept under wraps. For example, young athletes’ eagerness to be on an elite team and to compete at international levels may overcome their desire to speak out about what they have endured; others may fear that they will either not be believed or that they will be discriminated against for reporting such behaviors. Over the years, little improvement can be seen in child protection within sports, despite attempts to increase public attention, create relevant policies, and enforce immediate implementation of such acts and guidelines [7]. Moreover, sports organizations may even perpetuate the abuse of young athletes by sweeping such occurrences under the rug, conducting slow and private investigations, silencing victims, persuading athletes not to submit a report, and more [8]. Despite its detrimental outcomes, this phenomenon is greatly under-studied [9].

The term violence refers to a large range of behaviors that may differ greatly from one another, for example, in why it emerges, how it is expressed, and what its outcomes are Matthews and Channon [10]. Many definitions have been given to this concept, referring to violence as an attack or use of force that results in physical or psychological damage to those who are attacked, for example Audi [11], or a violation of people’s human rights or basic needs [12]. Matthews and Channon [10] distinguish between direct and indirect interpersonal violence, whereby the former relates to attempts to physically harm someone else’s body, while the latter relates to attempts to harm someone else’s property and assets. Such violence is intentional, intended, and expressive [13]. Direct violence can also be psychological rather than physical, causing severe emotional harm and distress [14]. In cases of physical violence, psychological harm is also inevitable. In 1991, Bourdieu [15] coined the term symbolic violence to describe a variety of situations in which people are excluded, treated in a derisive manner, or receive unequal treatment that leads to direct or indirect harm.

Studies on violence in sports tend to focus on a number of different themes, such as fan violence, that may occur before, during, and after sports events [16]; different types of violence in different types of sports [17]; and violence between rivals between and during competitions [18]. Yet one type of violence in sports that is much less addressed in the literature relates to violence exhibited towards athletes by coaches and other staff members, especially towards younger athletes [7]. One reason for this under-researched phenomenon is the difficulty to determine what is considered violent behavior in the coach-athlete relationship [19].

There is no dispute that coaches are required to help athletes achieve optimal performance. To do so, they must use a range of mental and physical methods, such as raising their voice or applying physical pressure. When used to a reasonable degree, such methods are perceived as legitimate and even necessary tools. For example, Jacobs et al. [20] found that coaches and team managers positively perceived a range of behaviors as an acceptable means for motivating and encouraging athletes—actions that may be perceived by others as less than desirable. A similar approach was seen in Stirling and Kerr [21], where one athlete who was interviewed said, “A tree is measured by its fruits, and if they reached their achievements in this way, there must be something right about it”. In other words, even athletes themselves may perceive such abuse as a necessary part of their training. This was also seen in Gervis et al. [22], who found that young athletes were more accepting of coaches’ violent behaviors towards children at high levels of sport. Yet, in contrast to this forgiving approach, other studies found that athletes reported emotional abuse that left them scarred even years after retiring. Kerr et al. [23], for example, interviewed eight female athletes from Canadian national sports teams of various disciplines, including Olympic athletes. In retrospect, the interviewees stated that the period in which they were athletes was the worst time of their lives. Even years after retiring, they still needed psychological therapy to help them deal with the emotional abuse that they had experienced.

A dearth of literature on coach violence in sports can be seen, with similarities and differences between countries and cultures. For example, when presenting athletes and coaches in the UK with a series of vignettes that depicted a coach’s emotionally abusive behavior towards a 14-year-old athlete, Gervis, Rhind, and Luzar [22] found that perceived competitive levels and athletic performance were negatively associated with perceived abusive coach behaviors. In other words, the higher the perceived athletic performance, the lower the perceived abuse. Additionally, McDonald and Kawai [24] examined perceptions of punishing coach behaviors among students from ten universities in Japan; these researchers found that participants normalized acts of coach violence and accepted them as necessary forms of discipline, often even interpreting such acts as indicative of coach care and kindness.

In the USA, Zogg, et al. [25] explored interpersonal violence in sports and perceived coaching styles among more than 4000 college athletes. The researchers found that almost 10% of the respondents reported having experienced at least one type of interpersonal violence during their college sports careers, even in the weeks immediately leading up to the study. Participants who reported having been exposed to interpersonal violence also reported worse psycho-social outcomes, including increased burnout, an expressed desire to quit their sport, and even decreased emotional well-being, especially among participants who identified as female and with non-heterosexual orientations.

In Portugal, Marracho et al. [26] asked young athletes about their experience or exposure to undesirable coach behaviors. As victims, no participants reported having experienced mistreatment behaviors in their athlete-coach relationship; yet as observers, many reported having witness various forms of abuse, including verbal and emotional abuse, as well as a lack of attention and support. Finally, in Sweden, Johansson and Lundqvist [27] explored sexual harassment and abuse, as well as coach–athlete relationships. The participants included more than 450 current and former sports-club athletes, of whom 5.5% reported some form of sexual harassment and abuse by their coach—mainly inappropriate, unpleasant, or offensive physical contact. No differences in coach harassment and abuse were seen between genders, sport performance levels, or individual/team sports. Moreover, 13–39% of the participants reported having experienced dependency, substantial coach influence over their personal-life, non-instructional physical contact, sexualized comments and jokes, and flirting. Such findings from different countries and regions suggest that coach behaviors and coach-athlete relationships are culturally rooted and may require major reconsideration of the role of sports in education. Comparative studies between countries could be especially beneficial in future research to further understand the culturally rooted issue of coach abuse.

Despite the relatively small degree of sports violence that has been reported [9], these can be categorized by a number of themes, such as research goals, gender differences, or types of violence. Psychological violence may include the restricting of movements, belittling, scapegoating, threatening, intimidating, discriminating, or ridiculing [28]; verbal violence may include profanities and humiliation [29,30,31], starvation or fattening [32,33,34], non-proportional punishing [30], and bullying, ignoring, or even expelling [35]; physical violence may include kicking, hitting, and beatings [9]; and finally, sexual violence may also occur against young athletes [36].

A human rights report that was recently published in Japan revealed vast violence by coaches towards athletes [37]. In Canada, this issue was placed on the public agenda following a series of revelations about the abuse of athletes between 1994 and 2005 (e.g., [23,37]). Recent years have seen an increase in awareness of the scope of the problem and its consequences [38]. In turn, increased efforts can be seen to gather global data as a means for equipping all parties involved in children and youth sports with relevant knowledge [5,39]. Moreover, acquiring such important data could enable the defining of rules, regulations, and codes of conduct, as well as the setting of ethical standards as a means for ensuring the physical and mental health and wellbeing of athletes.

Initially, the agenda for child protection in sports was driven by sexual abuse scandals [40], yet today it also addresses physical violence, psychological abuse, and even verbal hostility. While in the 1990s, few sports organizations acknowledged or addressed child abuse and protection (mainly in the UK, Canada, and Australia), an increase has been seen over the past decades in the international interest in this issue [40]; moreover, greater attempts have been made to describe the scope, characteristics, nature, and implications of this undesirable and harmful phenomenon [21].

In line with this literature review, this study presents four research objectives. First, based on the quantitative methodology approach, adults (aged 18+ years) who had been athletes as children and adolescents were asked to retrospectively report the types and frequencies of violence to which they were subjected by their coaches. Following their input, the research then attempted to differentiate between types of violence by their reported frequencies, while predicting which variables contribute to their perceived explanations for such violence. Next, based on the qualitative methodology approach, participants were asked to verbally describe the types of violence and difficulties that they had experienced as young athletes when faced with violence by their coaches. Following their input, the research then attempted to reveal how athletes perceive the consequences of such violence on their emotions, behaviors, and decisions, as both adolescents and as young adults.

## 2. Materials and Methods

The current study uses a Mixed Methods Phenomenological Research (MMPR) approach, specifically an exploratory sequential design. This means it starts with quantitative and continues with qualitative data to make the quantitative assessment more precise [41]. The reason for using MMPR is that phenomenological and quantitative methods can work well together under one main framework. Even though they have different beliefs about reality and knowledge, they share similar values and methods that make combining them possible [42].

Phenomenology, often considered a human science approach, shares scientific nature with deductive approaches, justifying the adoption of an MMPR approach. The flexibility of phenomenological methods, rooted in understanding subjective experiences, complements the strengths of quantitative methods. By incorporating multiple methods within a single study, researchers can enhance validity, minimize bias, and capitalize on the strengths of each method. The integration of quantitative methods can provide orientation, identify relevant phenomena, and test theories developed through phenomenological inquiry. Overall, the philosophical rationale for MMPR is grounded in the idea that combining phenomenological and quantitative methods under a single paradigm can yield a comprehensive and nuanced understanding of research phenomena [43]. Each section is presented separately in the paper.

### 2.1. Part 1—Quantitative Design

#### 2.1.1. Participants

The study included 440 participants (173 (39%) females), aged 27.6 (SD ± 8.48), who had trained for at least one full year in the youth department of a sports union in Israel. The participants’ average number of training years was 11.47 (SD ± 5.91) and their average age of retirement was 20.66 years (SD ± 7.61). They had an average of 5.3 coaches throughout their athletic career before the age of 18 (SD ± 4.37). Of the 440 participants, 164 had competed on an international level, 197 on a national level, and 64 on a country regional level, while 15 had not been competitive athletes. Among them, 204 (46.5%) had taken part in individual sports and 235 (53.5%) had taken part in team sports, as follows: 255 (60%) participated in ball games, 45 (10.6%) in martial arts, 56 (13.2%) in aquatic sports, 32 (7.5%) in track and field, and 37 (8.5%) in movement and dance. Finally, while most participants (302) had specialized in only one field of sport, 138 of the respondents had participated in more than one sport. Finally, 201 participants (46.1%) reported that their current field of occupation is related to sport, while 116 (26.6%) reported that it is not. The remaining 119 (27.3%) participants were still students at the time of the study, and as such did not yet have a clear field of occupation.

#### 2.1.2. Questionnaire

The aim of the questionnaire was to discover general trends of violence against boys and girls in child and youth sports departments. The questionnaire was developed and validated by Parent et al. [44]. In Parent et al. [44], exploratory structural equation modeling (ESEM) was used to identify latent factors that underly the various versions of the VTAQ. Their results showed that the VTAQ-Athlete includes nine items with three factors: psychological, physical, and sexual; The VTAQ-Coach includes 36 items with three factors: psychological/neglect, physical, and sexual, and the VTAQ-Parent includes 25 items with two factors: psychological/neglect and physical. We have translated it from French to Hebrew and back to French, using the back-translation approach (e.g., [45,46]).

While the original questionnaire included three sections (violence by friends, by coaches, and by parents), the current study only utilized the section on violence against athletes by their coaches. The questionnaire included 34 items that the participants were asked to rate on a Likert-like scale from 1 (never) to 4 (Very often more than 10 times), such as “Did the coach shake, push, grab, or throw you?” or “Did the coach throw an object at you?”.

#### 2.1.3. Procedure

The study was approved by the Institutional Review Board at the authors’ affiliated academic institution Kaye College for Education. To recruit participants, the questionnaire was sent to all sport unions and federations across Israel via e-mail, asking them to kindly forward it to their past and current athletes who are currently aged 18+ years. The same request was also sent to the Israeli Olympic Committee. In addition, the snowball method was applied via the authors’ personal social media pages.

The interviews, which were conducted via the Zoom platform, lasted 50–120 min and, with the participants’ approval, were recorded and transcribed. Upon completing the online questionnaire, the interviewees had stated that they would be willing to further discuss the issue of coach violence with the researchers. Our contact details (names, telephone numbers, and e-mail addresses) were included on the form. As with the questionnaire, complete confidentiality was ensured to all interviewees.

Analysis of the qualitative data was conducted by the first two authors and then reviewed by the additional two authors. The achieved input was analyzed and coded according to anticipated potential categories, based on the theoretical framework of the study, namely the types of violence. In other words, the data from the interviews underwent stages of categorization, including the describing, comparing, and relating of data to existing knowledge, as suggested by Attride-Stirling [47] and Bazeley [48]. Finally, trustworthiness was established using investigator triangulation [49], i.e., the participation of multiple researchers in the data interpretation process [50], as a means for enhancing clarification and accuracy.

For legal purposes, we chose to only include athletes over the age of 18; had we been exposed to violence against younger athletes, we would have had to report this to the police, according to the Child Protection Law that was enacted in Israel in 1995, and as such, would not have been able to maintain complete confidentiality for the respondents.

#### 2.1.4. Statistical Analysis

All data were analyzed using the SPSS software, version 18.0 (SPSS Inc., Chicago, IL, USA). Continuous variables are presented as mean ± SD, while categorical variables are presented as frequencies and percentages. Linear univariable analyses were performed to identify possible socio-demographic factors that are associated with coach violence scores. Multivariable linear regressions were performed forcing all covariates (socio-psychological and demographic variables) with a statistical significance of <0.175 into the model. Backward variable elimination was used to develop the regression model. Variables with a significance level of <0.1 were retained in the final model. The questions in the Violence Toward Athletes Questionnaire—VTAQ were divided into the following three categories: Psychological violence, physical violence, and sexual violence. The percentage of the scores for each category was calculated by dividing the reported score by the maximal score for the categories and then multiplying this number by 100. One-way analysis was performed to compare psychological, physical, and sexual violence with Bonferroni corrections. A *p*-value of <0.05 was considered significant.

### 2.2. Part 2—Qualitative Design

#### 2.2.1. Participants

This part of the research included 14 adults (9 females and 5 males), aged 23–37. They had all been competitive athletes for 10–15 years from the following branches of sports: surfing (1), gymnastics (2), judo (1), swimming (3), soccer (2), basketball (2), team handball (1), and tennis (2). While two had reached international competition levels, all others had stayed at the regional or national levels.

#### 2.2.2. Procedure

The interviews were conducted via Zoom and lasted 48–120 min. These interviews were initiated by the participants, who, after completing the questionnaire described above, had stated that they would be willing to further discuss the issue of coach violence with the researchers. Our contact details (names, telephone numbers, and e-mail addresses) were included. As with the questionnaire, complete confidentiality was ensured to all interviewees. A narrative approach [51] was applied by listening to the participants to as great an extent as possible while asking very few questions. The initial question of each interview was always: Tell me your story.

#### 2.2.3. Data Analysis and Trustworthiness

With the participants’ permission, the interviews were recorded and transcribed. Data analysis was then performed by the two first authors of this article and reviewed by the additional two authors. The input achieved was analyzed and coded according to anticipated potential categories based on the theoretical framework of the study, namely types of violence. The choice of three classifications suited the nature of the material, as reflected through the participants’ verbal expressions that were uttered during their interviews [52]. In other words, the data received from the athletes’ interviews underwent phases of categorization that included the describing, comparing, and relating of the data to existing knowledge, as suggested by Attride-Stirling [49] and Bazeley [48]. Finally, trustworthiness was established using investigator triangulation [49], i.e., the use of multiple researchers in interpreting the data [50] for clarification and accuracy.

## 3. Results

### 3.1. Part 1—Quantitative Design

The purpose of the qualitative section of the study was to examine how adults retrospectively perceive the extent of coach violence towards them as child or adolescent athletes. Table 1 presents the distribution of the questionnaire answers.

All participants experienced violence from their coach at least once during their career. Regarding physical violence, about 15% of the participants reported that their coach had shaken, pushed, or grabbed them; 12% reported that their coach had directly thrown an object at them; and 22% reported that they had been hit by an object that was not aimed directly towards them. In addition, about 28% reported that the coach had forced them or asked them to do extra high-intensity and excessive training until they were physically exhausted; 23% reported that they had been forced or asked to exercise while injured, despite having a medical opinion to the contrary; about 20% reported that they were forced or asked to perform movements or technical actions that were too difficult in relation to their abilities (physical or psychological) that could have or that actually had negative consequences on their health and safety. Psychological violence was more prevalent, with approximately 30% reporting that they had been yelled at, insulted, humiliated, and mocked; about half reported that they had been excessively criticized, expelled or suspended, ignored, or treated with indifference. Reports concerning sexual violence, however, were negligible.

The ratio of the questionnaire’s three types of violence were then calculated: psychological violence and neglect was most prevalent, followed by physical violence; sexual violence was the least reported factor (18.10 ± 4.22; 7.9 ± 2.02; 0.69 ± 0.17, respectively). One way ANOVA shows a significant difference between these three factors (*F*_(1,439)_ = 8453.002, *p* < 0.001), as seen in Figure 1.

When multivariable linear regressions were performed for all variables (age, gender, sport branch, individual/team sport, age of retirement from sports, number of training years, professional achievements, number of coaches, and place of residents), only two variables were significant predictors of violence: age of retirement and number of coaches, as seen in Table 2.

### 3.2. Part 2—Qualitative Design

After assessing the frequency of violent occurrences in the first part of the study, we were interested in understanding the types of violence and difficulties experienced by young athletes in greater detail. We also sought to reveal what they felt, how they coped, and how they perceive the consequences of such violence on their emotions, behaviors, and decisions in retrospect as today’s adults.

The thematic analysis that we conducted led to the emergence of six forms of coach-athlete violence. These themes are presented in the following sections together with quotes that support and expand on each of these themes: (1) psychological violence; (2) verbal violence; (3) starvation and food fattening; (4) non-proportional punishing; (5) physical violence; and (6) sexual violence. In addition to quotes that depict the violence to which the participants were subjected as young athletes, we have also added quotes that present the consequences of such violence as perceived by the participants today.

#### 3.2.1. Theme 1: Psychological Violence

This theme included restriction of movement, patterns of belittling, denigrating, scapegoating, threatening, scaring, discriminating, ridiculing, and other nonphysical forms of hostile treatment, such as rejection, expulsion, ignoring, bullying, or hazing. The following citations indicate psychological violence.

There were things that today as a grownup, I cannot comprehend how they were allowed to exist at all. For example, once I forgot to bring my bathing suit to practice. So she [the coach] said to me, ‘Okay, then swim in your underwear’. A 12-year old girl in a pool filled with people, open to the public. It doesn’t make sense to swim in your underwear, not even as a six-year-old girl. She said this in anger, ‘Swim in your underwear!’. There was no other option. She said, ‘Swim in your underwear or go home’. So I swam in my underwear.
But he would scream at me, terrorize me, swear at me. Later, there was a very, very strong issue of control. Who I could go out with, what I was allowed to say. It was like the ‘thoughts and feelings police’. If you lost a competition but didn’t really seem miserable—in his opinion—then he thought something was wrong with you. You should be sad because you lost and vice versa. If you wear certain clothes then it’s not appropriate, because it attracts too much attention. He brought me into sports at the age of eight, and we finished at the age of 30. We went through three Olympic Games, crises, and breakups. She ignored me every time I said something that didn’t meet her expectations. She actually expected us to be silent. She could ignore me for a whole week, like a punishment… So I learned to always keep my mouth shut.(Nurit)
He would scream so much, until he lost his voice, just because he was mad at something that we had done wrong.(Shir)
When we traveled, he kept my money. He would decide whether to give it to me or not. He used to threaten me and sometimes he even acted on his threats, and we couldn’t buy anything with our own money. In addition to expressing psychological violence, the interviewees spoke of how they coped with such violence, as seen in the following quotes.(Ori)
As a result, I lost my faith in her [the coach]. Training wasn’t fun anymore. But I kept on trying not to disappoint my parents. I retired with the feeling that I had dedicated my life to sports but had nothing to show for it… This was very difficult for me to let go of. I think you can see this even now as I talk about it, I’m still very emotional. I couldn’t talk about it for about a year. I wasn’t ready to talk about sports at all, at all… I also had issues with food, that was a very difficult relationship, I wanted to inhale everything without stopping to breathe. My parents started to worry about me. They would say, ‘You’re eating like a truck’… I simply replied that now that my coach isn’t there anymore, nobody can tell me what or what not to eat. I gained a lot of weight, and I was very upset that I had gained weight but then I didn’t know how to lose it. I was afraid of being hungry… I retired with the feeling that I was nobody, that I was a loser, that I wasn’t successful. I felt that I didn’t know who I was because I hadn’t succeeded. These feelings were with me all the time. Until I slowly started to recover and tried to escape from such associations as much as possible, from such people. The truth is that at first, I constantly needed feedback. Every time I did something, I needed someone to tell me if it’s okay or not. I had lost my self-esteem… There was no me. There was only a robot. I had to rebuild myself, and that was very, very difficult.(Tamar)

#### 3.2.2. Theme 2: Verbal Violence

This category of psychological violence included cursing and humiliation, as seen in the following citations.
Screaming like crazy at a young girl for minutes on end because she made a mistake?! That’s overreacting, right? And you would often hear crazy yelling. But you get used to it. You know you’ll be screamed at. You just don’t know when.(Tamar)

The following citation shows the outcome of this violence:
You say to yourself, it is what it is, there’s nothing I can do about it. Is there any other way to be on the national team and achieve your dreams? What can you do? He’s the coach. If you don’t want to [put up with it], you can stop. But if you want to move forward and make your dreams come true, then this is the only way.(Shir)
He knows how he behaves. It’s not a secret. Everyone knows it, everyone knows that he loses control and shouts and goes on a rampage and does illogical things… But it’s all part of the sport. He’s the coach. He wants you to succeed, he pushes you. So where’s the line? It’s really problematic.(Nave)
When I was 14, he would always laugh at me because of how I looked. He would tell me not to eat and that if I did eat, I should only eat salad. I was larger than my teammates. He used to call me: a pig, ball, fat, big butt. I couldn’t stand it. I felt humiliated. It was really unpleasant for me, I dedicated my life to swimming. And I really tried to improve, I came to all the training sessions, I never caused any problems. I always did as I was told. I even started running as well, and ate much less, but didn’t lose any weight. I was hungry all the time. Eventually I dropped out of the swimming team.(Noga)
She would humiliate me by imitating me crying. She was disrespectful towards my parents. She humiliated me in front of all the younger girls. But I did nothing. I just took it and took it. I didn’t enjoy the training sessions anymore. Once when a substitute coach arrived, I enjoyed myself so much that I suddenly remembered why I love gymnastics so much.(Tamar)
Whenever we saw him walking angrily down the corridor, no one wanted to even get close to him. No one wanted to accidentally see him. Think of how many hours, days, and weeks we spent with him… and when the atmosphere is so aggressive, you start to feel ill. You carried the fear with you.
He insisted that our lives be devoted to sports. No family distractions, no boyfriends, no work, no school. We had to be sport nuns…(Yael)

#### 3.2.3. Theme 3: Starvation and Food Fattening

This theme included weight cycling practices and rapid weight loss, as seen in the following quotes.
She would weigh me four times a day. Before training, after the first training, before the second training, and after the second training. Every single time. One day, the goal was to reach 48 Kg. But if I had reached 48, she would have demanded that I reach 47. It would never end. One day, I weighed 50 kg I think at the beginning of training, so I was supposed to weigh 47 by the end. You have to understand how thin I was… But she just said: ‘You don’t start training until you’re 47 kg, I don’t care how, do whatever you want, hang me, take pills, throw up. I don’t care what you do, you’re not starting until you’re 47 kg. I’m lucky I don’t like to throw up. I said OK. She told me to start running, even though I have stress fractures. I started running, I ran for like an hour. I lost one kilo. Then I only drank water. No food. Just water. But I couldn’t keep running anymore. I was on a treadmill so I started walking. I said to myself, at least I’ll walk, I can’t run. It wasn’t like I was running with a T-shirt and leggings. You need to understand how I was dressed. I was wearing long pants, a short-sleeved shirt, a long-sleeved shirt, a lined jacket, and a coat when she came to check on me. But when she saw that I was walking and not running, she started screaming and swearing at me, like she had completely lost control. She shouted at me to start jumping next to her with the jump rope. So for the next two-and-a-half hours, I jumped rope next to her. I was not allowed to stop!!! My legs were on fire. I couldn’t feel them anymore. It hurt like crazy. After that, she told the personal trainer to take me to run in the sand dunes. It was like that almost every evening. He weighed me and I was 47.7 kg. He was so kind, he just said, ‘Well, without your clothes your weight is OK’.I swear I hadn’t eaten, I wasn’t allowed to drink. She wouldn’t let me drink. She said it makes my muscles swell and makes me heavy.Needed to see me sweat, huge amounts of sweat. So what did I do? I wouldn’t drink all day and then I would come to the room at the end of training and drink 3 L. But then I would feel really heavy, I was gaining 2–3 kg just from the water I drank, and then the next morning, she would weigh me again and ask why I had eaten the night before. She would shout at me that I’m a liar, a thief. And I would try to tell her that I’ve only been drinking, not eating, and then she would scream even more, saying, ‘But I told you not to drink!’. How was I supposed to exist if I couldn’t drink during the day, couldn’t drink at lunchtime, couldn’t drink in the evening, couldn’t drink at night…?At first, I tried really hard to do what she told me to do. She was my mentor, she knows what’s right… I really did what she told me, but it was just impossible to function like that. Absolutely impossible. At breakfast, she would give me one cucumber and an egg. Sometimes I had lunch, sometimes not, depending on my weight. If I was given lunch, it was a salad or even just an apple, and then in the evening… vegetables and maybe some yogurt. I was always hungry. I would think about food all the time, even while training.(Noa)

The following quotes exemplify the consequences of such violence:
For years I suffered from eating disorders and from a distorted body image.I gained weight and then had no clue how to lose it…(Noa)

#### 3.2.4. Theme 4: Non-Proportional Punishing

One of the themes that emerged from the qualitative study was non-proportional punishing, as seen in the following quotes.
My friend was fooling around before the game. The coach got angry and threw him off the field. I begged him to punish him after the game. He was our leading player. But he just screamed at me too and told me to get off the field. I didn’t play that day. I had been preparing for this game for months, my parents had come especially to see me play. I sat outside feeling very humiliated.(Ofek)
In general, it’s humiliating when you’re a mature person and you’re being yelled at in front of other people, in front of the other competitors. There were loads of comments all the time. Even about other teammates. Like the pants that she was wearing that had to many frills. He [the coach] thought they were attracting too much attention, so he said to her, ‘If you want attention, win some medals. But don’t wear those pants anymore’. Then one day she wore them on purpose, and he just dropped her off in the middle of nowhere while we were on the way to the airport in Berlin! She had to make her own way to the airport!(Ofek)

In response to such non-proportional punishing, two interviewees expressed the following outcomes:
When I [later] became a coach myself, I knew how *not* to behave towards my young athletes.(Yam)
At the time, we were very obedient. No one wanted to be punished or thrown off the team. We knew that we depended on him, and he was considered the best.(Gonen)

#### 3.2.5. Theme 5: Physical Violence

This theme included kicking, punching, and beatings, as seen in the following quotes.
I just didn’t do the entrance of the choreography the way she wanted me to. So she gets up, tugs on my arm, and shouts, ‘What did you do?!’. And then she steps on me. She knew I had a stress fracture… I’d had it for about two years, but she stood on me!… There were other little girls in the gymnasium… It was really unpleasant. Then suddenly something occurred to me, and I answered her! For the first time ever, I answered her. I was really angry and I said, ‘Enough is enough!’. She was so shocked that I had even reacted. But then she looked at me and yelled, ‘What did you say?!’.(Tamar)
She took me by the hand just like that and then pushed me. She really shoved me and I fell. And then she started kicking me, and shouted, ‘Get out of here!’. I was so humiliated, so, so humiliated. All the little girls were watching… I just got up, took my bag, and left.
All the time I felt like I wasn’t OK. No matter what I did. During training, the coach would swear at us, shout and humiliate us. He didn’t care that I was in pain. For example, I had stress fractures in my legs. But he thought I was inventing my pain. He would say that I was making it up just so that I wouldn’t have to make an effort. But he knew that I had a stress fracture in my leg. I told him that my leg was hurting. One day I really couldn’t jump because of the pain. When I told him, he asked where the pain was. I thought he was asking because he cared. But he just started standing on my leg—really hard, again and again! He told me that I need to go beyond the pain. And I was really trying to overcome the pain and not think about it.(Ofek)
You could have some kind of injury. A tear, a torn or stretched ligament, or something like that. You know it takes six weeks to heal, let’s say, but if you have a competition or training camp coming up, then you go to compete or train even with your injury. Instead of letting it heal properly. Because you understand that if you don’t, you’ll be out of action for much longer than the six weeks. Because you’ve already missed some competitions that were the basis for further competitions. So you compete like this.(Nir)
Once she got mad at me for being sick, I really didn’t feel well. We were at a training camp and the pool was really cold, I was running a fever of 40. So she told me to get into the water to cool down. She said that I have to take part in the training.(Shahar)

In relation to physical coach-athlete violence, the following outcomes were also conveyed in the interviews:
Unless you’re dead or dying… even if you’re really ill, you just stay quiet. Otherwise they’ll say that you’re whining. Even though it’s better to miss one training session, rather than killing your body, but I’m like… I’m telling you, everyone around me, myself included, will come to train even if we’re really ill. Everyone’s afraid to say that they’re not coming.(Ofek)

#### 3.2.6. Theme 6: Sexual Violence

This theme included behavior of sever harassment, exploitation, and abuse as described by Fejgin and Hanegby [53].
We were in the hot tub, like, after training we usually go into the hot tub, and he [the coach’s son], held on to me and took his penis out. At first I thought it was his toe. I was an innocent 14-year-old. I didn’t know what it was, I’d never seen anything like it in my life. When I eventually realized what it was, I froze, I froze in my place. I couldn’t do anything, I just froze… One of the other girls said that he used to touch her, and suddenly all the girls were sitting down with us, talking about it. He’d done that to many of the girls. It was very comforting to know that we weren’t alone. But we didn’t do anything about it. Nothing.(Noga)
From time to time he would touch me, more and more. It got steadily worse. It started with him touching my genitals. Then he would ask me to play something that requires concentration, and I would be really concentrating on the game with the remote control. And he would start to touch me. Then he would pull me on top of him. At first he was fully clothed, but then very gradually, very gently, he would take his pants off. He would also put cream on. The last time it happened, he took me to his room… and lay me on the bed, and started rubbing himself against me. He almost penetrated… I have a picture in my head of some I of mirror on the wall, and I can see myself crying…(Nave)

For this sixth and final theme, interviewees also spoke of the outcome of their being subjected to sexual abuse, as seen in the following quotes.
I didn’t talk to anyone [about it]. I was so ashamed. I was even ashamed to tell my parents. I was really ashamed, because somehow, I felt like I was guilty. Why? I felt that I was guilty, like, why had I gone into the jacuzzi alone with him? Or what did I expect wearing a bathing suit?(Yonit)
When I talked to my teammates about it, everyone thought he had a kind of deviant profile. You know, like if someone brings something up, then everyone immediately says his name or remembers how he used to behave. They all remember this. If he had been a responsible adult, this wouldn’t have happened. It wasn’t really a secret though. So it should have turned on a warning light for other people in the pool. This stuff went on for about a year.(Shir)
We always have this talk, my friends and I, about the babysitter who looks after my kids. I never let my husband take her home, I always take her home myself. Not because I think my husband will do something to her, but because I don’t want there to be a situation where she may not feel comfortable, or that she might say something that happened or didn’t happen. My friends and I always talk about where the line should be drawn.(Nurit)
When you send a child to a class, you know a certain person or you think you know a certain person. But then it’s someone completely different at the core. That’s how it was with my coach… Today I have serious trust issues and it’s clear to me that it’s based on something from there. I do work on these things, but my first instinct is not to trust the person, not to believe the person who is standing in front of me…(Ori)
I felt that I was like… I was very careful to hide everything… but I felt that… When I was there, it seemed funny and I laughed about it. But then at home, I would sit and think about these things, about how to get out of this situation… I felt like I was living a sort of double life with some kind of mask that I put on in the morning and only took off at night… There were many things that never even crossed my mind as being problematic until after I stopped training. Even then I repressed many things and I am only able to see today. I might suddenly remember a certain situation… It’s really difficult for me.(Yam)
I don’t remember getting out of the car [after training]… I just remember how I would feel five seconds later, because I would just sit on the steps at the entrance to my house and cry. Eventually I would collect myself, wipe away the tears, and go into the house as if nothing had happened. Then I would take a shower, and I remember scrubbing my body really hard. I felt like I really needed to clean myself. Even later, when I started dealing with these things as a more mature person, I would scrub my skin until it hurt whenever I was in a difficult situation.(Noga)
I was afraid to talk about it. I was afraid, I didn’t know… like if I look back on it today, I don’t know if I was afraid of my parents’ reaction, or if I was afraid for my place, or if I was afraid that he would deny everything and I would look like a liar… I don’t know what I was thinking, why I didn’t contact anyone, but I really remember trying to get my family to ask me what had happened or how I felt.

## 4. Discussion

### 4.1. Part 1—Quantitative Design

The aim of the quantitative section of this study was to examine the types and frequencies of violence experienced by young athletes by their coaches, as reported retrospectively by adult athletes (18 years+), to differentiate between types of violence by their reported frequencies, and to predict which variables contribute to their perceived explanations of coach violence towards young athletes. Unlike previous reports (e.g., [23,37,38]), our findings show a relatively low frequency of violence in all three factors (psychological violence, physical violence, and sexual violence).

Specifically, sexual violence was rarely reported in this study. There may be several possible explanations for these results. First, in Israel, there are laws for preventing sexual harassment [54,55]. All places of occupation in Israel are obliged to nominate a specific person who is in charge of sexual harassment in the workplace. Sexual violence is treated as a criminal act and handled by the police. Hence, public awareness towards sexual harassment has significantly increased over the past two decades. On the other hand, the underreporting of sexual harassment is a known phenomenon [55] that might have occurred in this study. With regards to physical violence, the majority of the participants reported having never experienced any of the physical behaviors that were measured in this study. Nevertheless, a “relatively” low percentage of participants did report having experienced physical violence. The question that every coach, parent, and adult that works or lives with children and youth should ask themselves is, what is “relatively”? Is it acceptable that 20% or 15% of young athletes experience physical violence by their coaches? Should we be concerned if “only” 10% experience physical violence, or should we emphasize that 90% do not?

As for psychological violence and neglect, the results are not encouraging. It seems that almost all participants experienced psychological violence to some extent. The variety of offensive/insulting/harmful/humiliating behaviors that coaches use towards their athletes is also troublesome, as athletes spend hours, days, months, and sometimes even years in the company of their coaches [56,57]. The participants expressed having experienced endless negative emotions, such as frustration, fatigue, lack of motivation, boredom, shame, and anxiety—feelings that were probably heightened by the psychological violence that they experienced. The findings of this study should ring a warning bell for all parties involved in the education and sports training of children and adolescents.

Lastly, the regression analysis that we conducted revealed that the number of coaches and the age of retirement are the only variables that predict the athletes’ having experienced violence by their coaches. In other words, the older they were when retiring from their sports, the more likely they were to experience such violence. This could be explained by the parents’ decreased involvement as the age of the child increases. In other words, as children age, they become less dependent on their parents. Hence the role of peers and coaches may become more prominent, as does the training environment climate [58,59]. In addition, as young athletes age, they become more competitive and the odds of their experiencing violence increase [60,61].

These findings and interpretations are important and add to the field of literature on coach violence towards athletes. Yet several limitations should be considered. First, the input received from the interviewees was based on retrospective self-reporting that is highly dependent on memories, which tend to fluctuate over time. Nevertheless, a similar method has been used in previous studies (e.g., [21]). Second, the sample in this study was not random but rather one of convenience. Yet this limitation is overcome by the relatively large sample size that presents a variety of voices.

Following this quantitative section of the study, a qualitative research design was also applied to uncover the subjective meanings and long-term effects of the violence to which the participants were exposed as young athletes.

### 4.2. Part 2—Qualitative Design

By analyzing the interviews conducted in this study, we were able to address our research objectives. First, we identified six forms of coaches’ violent behaviors towards athletes, in line with previous studies: (1) psychological violence [28]; (2) verbal violence [29,30]; (3) starvation [33,34]; (4) non-proportional punishing [30]; (5) physical violence [9]; and (6) sexual violence [36]. Adding important information to the literature, the study presents each form of coach-athlete violence together with detailed examples and related outcomes. These reports were provided by adults who found it difficult to recall what they had been through and put their experiences into words. Even years later, some participants were afraid to be exposed through their stories.

Stirling and Kerr [21] claimed that violence affects the psychological wellbeing, training, and performance of athletes, and as such, they may exhibit frequent mood swings, anger, low self-efficacy, low self-esteem, anxiety, and even lack of a sense of accomplishment. They may also exhibit decreased motivation, reduced enjoyment during training, impaired focus, and difficulty in acquiring skills. Some athletes spoke of friends who had retired early from sports since they could not cope with their coach’s violent behavior towards them. Finally, such abuse may lead to decreased levels of performance and inability to reach their athletic potential. Indeed, during our interviews, violence was described as having a far-reaching effect, beyond the scope and time of their actual career.

While some athletes stated that their coaches’ constant yelling at them actually spurred them on, making them better athletes, they would not willingly go through such abuse again. Our results strengthen those of Kavanagh et al. [62], who examined retrospective reports of retired elite athletes in England. As with this study, their participants also expressed the difficult experiences that they had undergone subject to coach-athlete abuse, as well as the coping strategies that they employed in order to persist and achieve their goals. The athletes spoke of different ways in which they dealt with isolated or ongoing incidents of abuse over the years, and how they dealt with the memories and consequences of this abuse following their retirement.

Despite the important contribution of the qualitative research presented in this article, several research limitations should be addressed concerning the qualitative study. First, as the interviews employed a retrospective study, memory-related shortcomings and emotional changes may have occurred over time—for better or for worse. However, such a methodology is greatly accepted in academic research [63]. Moreover, only 14 of the 440 participants agreed to talk about their experiences in an interview. While this may be a relatively small and non-representative sample, the stories that were conveyed by these interviewees are alarming enough to call for immediate practical implications and interventions.

## 5. Conclusions

When combining both parts of this study—the quantitative and the qualitative—the findings indicate that athlete-coach abuse is alive and kicking. As such, it is imperative to increase public awareness on this matter, introduce the topic of young athletes’ safety into coaching education, and nominate position holders to be in charge of such safety within sports organizations. The novelty of this study’s results is primarily expressed through exploration of the lingering and ongoing impact of coaches’ violence and abuse against athletes. Furthermore, the employed methodology, which integrated both quantitative and qualitative research, enabled the participants to sound their voice and put a so-called face to the numbers. This research approach is relatively rare in studies on violence in sports. As such, the study’s significance lies in its ability to humanize the individuals behind these incidents, particularly children.

## Figures and Tables

**Figure 1 behavsci-14-00486-f001:**
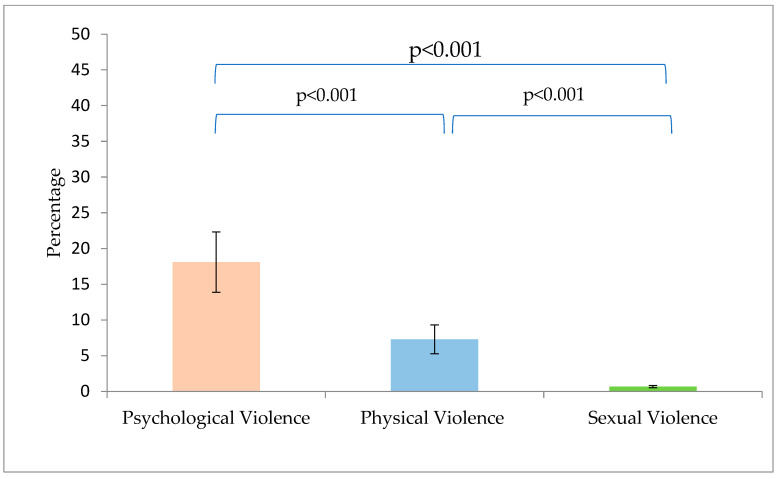
Differences Between the Questionnaire’s Factor Scores.

**Table 1 behavsci-14-00486-t001:** Distribution of the Questionnaire Answers in Percentages (*n* = 440).

	Items	1 Never	2 Rarely (1–2 Times)	3 Some-Times (3–10 Times)	4 Very Often (>10 Times)
1	Shook, pushed, grabbed, or threw you	84.5	11.2	2.3	2
2	Threw an object directly at you	88.2	8.1	3.0	0.7
3	Hit you with the hand (for example, slaps)	92.0	4.1	3.0	0.9
4	Punched or kicked you	97.0	1.1	1.6	0.3
5	Hit you with an object (for example, sports equipment)	91.1	6.9	1.1	0.9
6	Tried to strangle you	98.0	0.4	1.4	0.2
7	Hit or threw objects that were not aimed directly at you (e.g., water bottles, pens)	78.0	12.0	8.0	2.0
8	Forced you/instructed you to injure an opposing player	90.5	6.4	2.0	1.1
9	Forced you/instructed you to humiliate or mock an opponent	92.0	6.1	1.4	0.5
10	Forced you/instructed you to threaten or hurt an opponent	92.0	5.0	2.3	0.7
11	Allowed you to injure an opposing player (with a punch, sports equipment, etc.) in a competition, without intervening	93.6	5.2	1.2	/
12	Allowed you to humiliate or mock an opponent in a competition, without interfering	90.7	6.1	2.3	0.9
13	Allowed you to threaten or hurt an opponent in a competition without intervening	91.8	4.5	3.0	0.7
14	Threatened to leave/abandon you	88.9	5.5	4.3	1.3
15	Threatened to harm you	91.4	3.6	4.3	0.7
16	Threatened to harm someone or something you love	95.2	3.4	1.4	/
17	Yelled at you and insulted you, humiliated you, and mocked you	59.5	23.0	11.4	6.1
18	Criticized you excessively (e.g., about your performance or your attitude)	50.2	23.6	18.9	7.3
19	Expelled or suspended you	72.5	20.9	4.6	2.0
20	Locked you in a closed room or tried to limit your freedom of movement (e.g., locking you in the locker room, tying you up)	97.7	1.1	1.1	/
21	Asked you to limit or reduce your social connections (with friends, romantic partners, or family member) to enable you to invest more of yourself in your sports	77.0	15.9	5.9	1.1
22	Ignored you or treated you with indifference (e.g., refused to talk to you, ignored your existence)	66.6	20.5	9.1	3.9
23	Forced you/instructed you to do extra high-intensity and excessive training until you were exhausted	72.5	15.2	8.9	3.4
24	Forced you/instructed you to exercise while injured despite having a medical opinion to the contrary	77.0	13.0	6.6	3.4
25	Forced you/instructed you to perform movements or technical actions that are more difficult than you are capable of (physically or psychologically) that had or could have had negative consequences on your health and safety	78.2	16.4	3.4	2.0
26	Asked you to use prohibited substances to reach the desired weight for the sport (fasting, vomiting, pills)	96.8	1.6	1.1	0.5
27	Asked you to use prohibited substances to improve performance (steroids, hormones)	97.1	1.1	0.2	/
28	Knew that you had used prohibited substances to reach the desired weight for the industry	98.2	0.5	1.1	0.2
29	Knew that you had used prohibited substances to improve performance	98.6	0.5	0.9	/
30	Asked you to stop going to school or suspend your studies in order to devote yourself to sports	88.2	7.7	2.0	2.0
31	Made rude, insulting comments that made you uncomfortable about your sex life, your private life, or your physical appearance (for example, comments about you or your partner’s intimate body parts)	87.3	8.6	2.7	1.4
32	Behaved sexually in a way that made you feel uncomfortable (for example, rubbing you, staring at you, undressing you with their eyes, whistling at you, and massaging you)	88.2	8.4	1.7	1.7
33	Watched you or force you to perform a sexual act (touching yourself, themselves, or others)	94.3	3.9	1.4	0.5
34	Photographed you while you were having sexual activity (touching yourself, themselves, or others)	97.7	1.4	0.7	0.2

**Table 2 behavsci-14-00486-t002:** Predictor of VTAQ Score Using Linear Regression Analysis.

Variable	Standardized β (CI)	*t*	*p*
Age	−0.05 (−0.15–0.04)	−1.08	0.27
Gender	0.08 (−0.25–3.22)	1.67	0.94
Sport type	−0.13 (−0.75–0.56)	−0.27	0.78
Individual/team	0.23 (−1.30–2.16)	0.48	0.62
Age of retirement	0.12 (0.01–0.257)	2.15	0.032 *
Retirement: yes/no	−0.06 (−2.97–0.59)	−1.30	0.19
Achievements	−0.01 (−1.16–0.92)	−0.22	0.81
Years of practice	0.09 (−0.003–0.29)	1.926	0.055
Number of coaches	0.14 (0.108–0.496)	3.056	0.002 *
Occupation	0.031 (−0.787–1.575)	0.655	0.513
Residence	−0.012(−1.717–1.342)	−0.241	0.810
Zone	0.029 (−0.386–0.716)	0.0589	0.0556

* = *p* < 0.5.

## Data Availability

Data is unavailable due to privacy and ethical restrictions.

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
