# Peer review of "Clear Yet Crossed: Athletes’ Retrospective Reports of Coach Violence"

_behavsci, 2024, doi:10.3390/bs14060486_

Round 1

Reviewer 1 Report

Comments and Suggestions for Authors

Dear authors,

The paper addresses a very critical topic on coach-athlete violence, highlighting its significance in public discourse.

The introduction accurately contextualizes the study, emphasizing its importance and outlining the purpose of the work, with content succinctly described and contextualized with respect to theoretical background and empirical research on the topic. The significance of the research is clearly defined. The Research Methodology is thorough, providing detailed descriptions of the mix-methods design approach, however some points should be attended. Although conclusions are supported by the findings, they could benefit from more thoroughness given the importance of the topic. In my view, the manuscript is suitable for publication in Behavioral Sciences, pending minor revisions.

Please consider the following concerns:

Regarding formatting, there are inconsistencies in font types, and Figure 1 may not adhere to the journal's guidelines. Additionally, Table 2 as also a different font/colour.

L. 158: “The questionnaire was developed and validated by Parent et al. (2019), and translated from French to Hebrew and back to French, using the back-translation approach (e.g., Brislin, 1970; Klotz et al., 2022)”. – what type of validation was conducted?

L. 191 – “Part 2—Qualitative Design” - how was the sample collected? Was it random or convenience sampling? Were consent forms obtained and signed?

L. 571 - Limitations of the qualitative study should also be addressed.

In my opinion, since this is a vital topic, authors should carefully highlight the implications of the study.

Author Response

Reviewer 1:

The paper addresses a very critical topic on coach-athlete violence, highlighting its significance in public discourse.

The introduction accurately contextualizes the study, emphasizing its importance and outlining the purpose of the work, with content succinctly described and contextualized with respect to theoretical background and empirical research on the topic. The significance of the research is clearly defined. The Research Methodology is thorough, providing detailed descriptions of the mix-methods design approach, however some points should be attended. Although conclusions are supported by the findings, they could benefit from more thoroughness given the importance of the topic. In my view, the manuscript is suitable for publication in Behavioral Sciences, pending minor revisions.

Thank you for your positive feedback. We have implemented the changes that you suggested – in the revised manuscript and in response to each of your comments, highlighted below in yellow:

Please consider the following concerns:

  1. Regarding formatting, there are inconsistencies in font types, and Figure 1 may not adhere to the journal's guidelines. Additionally, Table 2 as also a different font/colour.

Thank you for pointing this out. We have amended both the figure and the table.

  1. L. 158: “The questionnaire was developed and validated by Parent et al. (2019), and translated from French to Hebrew and back to French, using the back-translation approach (e.g., Brislin, 1970; Klotz et al., 2022)”. – what type of validation was conducted?

We have added the following text:

In Parent et al. (2019), exploratory structural equation modeling (ESEM) was used to identify latent factors that underly the various versions of the VTAQ. Their results showed that the VTAQ-Athlete includes nine items with three factors: psychological, physical, and sexual; The VTAQ-Coach includes 36 items with three factors: psychological/neglect, physical, and sexual, and the VTAQ-Parent includes 25 items with two factors: psychological/neglect and physical. 

  1. L. 191 – “Part 2—Qualitative Design” - how was the sample collected? Was it random or convenience sampling? Were consent forms obtained and signed?

Thank you for highlighting this oversight. The revised manuscript now reads:

The study was approved by the Institutional Review Board at the authors’ affiliated academic institution [name removed for masked review]. To recruit participants, the questionnaire was sent to all sport unions and federations across Israel via e-mail, asking them to kindly forward it to their past and current athletes who are currently aged 18+ years. The same request was also sent to the Israeli Olympic Committee. In addition, the snowball method was applied via the authors’ personal social-media pages.

The interviews, which were conducted via the Zoom platform, lasted 50–120 minutes and with the participants’ approval were recorded and transcribed. Upon completing the online questionnaire, the interviewees had stated that they would be willing to further discuss the issue of coach violence with the researchers. Our contact details (names, telephone numbers, and e-mail addresses) were included on the form. As with the questionnaire, complete confidentiality was ensured to all interviewees.

Analysis of the qualitative data was conducted by the first two authors, and then reviewed by the additional two authors. The achieved input was analyzed and coded according to anticipated potential categories, based on the theoretical framework of the study – namely the types of violence. In other words, the data from the interviews underwent stages of categorization, including the describing, comparing, and relating of data to existing knowledge, as suggested by Attride-Stirling (2001) and Bazeley (2009). Finally, trustworthiness was established using investigator triangulation (Anney, 2014), i.e., the participation of multiple researchers in the data interpretation process (Cho & Trent, 2006) – as a means for enhancing clarification and accuracy.

  1. L. 571 - Limitations of the qualitative study should also be addressed.

We have now added the following limitations regarding the qualitative study:

Despite the important contribution of the qualitative research presented in this article, several research limitations should be addressed concerning the qualitative study. First, as the interviews employed a retrospective study, memory-related shortcomings and emotional changes may have occurred over time – for better or for worse. However, such a methodology is greatly accepted in academic research (Dai and Li, 2020). Moreover, only 14 of the 440 participants agreed to talk about their experiences in an interview. Yet while this may be a relatively small and non-representative sample, the stories that were conveyed by these interviewees are alarming enough to call for immediate practical implications and interventions.

The following references have been added to the revised manuscript:

Dai, D.Y.; Li, X. Behind an Accelerated Scientific Research Career: Dynamic Interplay of Endogenous and Exogenous Forces in Talent Development. Educ. Sci. 202010, 220. https://doi.org/10.3390/educsci10090220

Johansson, S., & Lundqvist, C. (2017). Sexual harassment and abuse in coach–athlete relationships in Sweden. European Journal for Sport and Society14(2), 117–137.‏

Marracho, P., Coelho, E. M. R. T. D. C., Pereira, A., & Nery, M. (2023). Mistreatment behaviours in the Athlete-coach relationship. Retos, 2023(50), 790–7989.‏

McDonald, B., & Kawai, K. (2017, April). Punishing coaching: Bukatsudō and the normalisation of coach violence. Japan Forum, 29(2), 196–217.

Zogg, C. K., Runquist, E. B., Amick, M., Gilmer, G., Milroy, J. J., Wyrick, D. L., ... & Tuakli-Wosornu, Y. A. (2024). Experiences of interpersonal violence in sport and perceived coaching style among college athletes. JAMA Network Open7(1), e2350248–e2350248.‏

Reviewer 2 Report

Comments and Suggestions for Authors

The authors present a compelling exploration into the phenomenon of coach-athlete violence through retrospective reports of adults who were athletes during their formative years. The study's aims are well-articulated, focusing on examining the prevalence, predictors, difficulties faced, and outcomes of coach violence on athletes' emotions and behaviors. The methodology, comprising both quantitative self-reporting questionnaires and qualitative interviews, is commendable for its comprehensiveness in capturing the multifaceted nature of coach-athlete dynamics. This article is of a high standard, demonstrating exemplary methodology, and holds significant societal implications.

However, it would benefit from discussing existing research on cultural variations in attitudes towards coach violence. If such studies exist, incorporating them into the introduction would enhance the contextual understanding of the issue. Alternatively, if there is a dearth of literature in this area, suggesting the need for future research in cross-cultural comparisons would be prudent.

Furthermore, the article lacks explicit information regarding participant informed consent procedures, privacy protection, confidentiality measures, and ethical equality considerations, as recommended by ethical guidelines such as the Declaration of Helsinki and the Publication Manual of the American Psychological Association. Clarity on these ethical aspects would strengthen the study's methodological rigor.

I commend the authors for conducting this important study and anticipate its significant impact on the field. I am confident that the article will garner substantial attention and citation in the future.

Author Response

Reviewer 2:

In my opinion, since this is a vital topic, authors should carefully highlight the implications of the study.

  1. The authors present a compelling exploration into the phenomenon of coach-athlete violence through retrospective reports of adults who were athletes during their formative years. The study's aims are well-articulated, focusing on examining the prevalence, predictors, difficulties faced, and outcomes of coach violence on athletes' emotions and behaviors. The methodology, comprising both quantitative self-reporting questionnaires and qualitative interviews, is commendable for its comprehensiveness in capturing the multifaceted nature of coach-athlete dynamics. This article is of a high standard, demonstrating exemplary methodology, and holds significant societal implications.

Thank you so much for your positive input.

  1. However, it would benefit from discussing existing research on cultural variations in attitudes towards coach violence. If such studies exist, incorporating them into the introduction would enhance the contextual understanding of the issue. Alternatively, if there is a dearth of literature in this area, suggesting the need for future research in cross-cultural comparisons would be prudent.

Following this important feedback, we have now added the following paragraphs to the revised manuscript:

A dearth of literature on coach violence in sports can be seen, with similarities and differences between countries and cultures. For example, when presenting athletes and coaches in the UK with a series of vignettes that depicted a coach’s emotionally abusive behavior towards a 14-year-old athlete, Gervis, Rhind, and Luzar (2016) found that perceived competitive levels and athletic performance were negatively associated with perceived abusive coach behaviors. In other words, the higher the perceived athletic performance, the lower the perceived abuse. Additionally, McDonald and Kawai (2017) examined perceptions of punishing coach behaviors among students from ten universities in Japan; these researchers found that participants normalized acts of coach violence and accepted them as necessary forms of discipline – often even interpreting such acts as indicative of coach care and kindness.

In the USA, Zogg, et al. (2024) explored interpersonal violence in sports and perceived coaching styles among more than 4,000 college athletes. The researchers found that almost 10% of the respondents reported having experienced at least one type of interpersonal violence during their college sports careers – even in the weeks immediately leading up to the study. Participants who reported having been exposed to interpersonal violence also reported worse psycho-social outcomes, including increased burnout, an expressed desire to quit their sport, and even decreased emotional well-being – especially among participants who identified as female and with non-heterosexual orientations.

In Portugal, Marracho et al. (2023) asked young athletes about their experience or exposure to undesirable coach behaviors. As victims, no participants reported having experienced mistreatment behaviors in their athlete-coach relationship; yet as observers, many reported having witness various forms of abuse, including verbal and emotional abuse, as well as a lack of attention and support. Finally, in Sweden, Johansson and Lundqvist (2017) explored sexual harassment and abuse, as well as coach–athlete relationships. The participants included more than 450 current and former sports-club athletes, of whom 5.5% reported some form of sexual harassment and abuse by their coach – mainly inappropriate, unpleasant, or offensive physical contact. No differences in coach harassment and abuse were seen between genders, sport performance levels, or individual/team sports. Moreover, 13–39% of the participants reported their having experienced dependency, substantial coach influence over their personal-life, non-instructional physical contact, sexualized comments and jokes, and flirting. Such findings from different countries and regions suggest that coach behaviors and coach-athlete relationships are culturally rooted, and may require major reconsideration of the role of sports in education. between countries could be especially beneficial in future research, to further understand the culturally rooted issue of coach abuse.

  1. Furthermore, the article lacks explicit information regarding participant informed consent procedures, privacy protection, confidentiality measures, and ethical equality considerations, as recommended by ethical guidelines such as the Declaration of Helsinki and the Publication Manual of the American Psychological Association. Clarity on these ethical aspects would strengthen the study's methodological rigor.

Thank you for highlighting this issue. The revised manuscript now reads:

The study was approved by the Institutional Review Board at the authors’ affiliated academic institution [name removed for masked review]. To recruit participants, the questionnaire was sent to all sport unions and federations across Israel via e-mail, asking them to kindly forward it to their past and current athletes who are currently aged 18+ years. The same request was also sent to the Israeli Olympic Committee. In addition, the snowball method was applied via the authors’ personal social-media pages.

The interviews, which were conducted via the Zoom platform, lasted 50–120 minutes and with the participants’ approval were recorded and transcribed. Upon completing the online questionnaire, the interviewees had stated that they would be willing to further discuss the issue of coach violence with the researchers. Our contact details (names, telephone numbers, and e-mail addresses) were included on the form. As with the questionnaire, complete confidentiality was ensured to all interviewees.

Analysis of the qualitative data was conducted by the first two authors, and then reviewed by the additional two authors. The achieved input was analyzed and coded according to anticipated potential categories, based on the theoretical framework of the study – namely the types of violence. In other words, the data from the interviews underwent stages of categorization, including the describing, comparing, and relating of data to existing knowledge, as suggested by Attride-Stirling (2001) and Bazeley (2009). Finally, trustworthiness was established using investigator triangulation (Anney, 2014), i.e., the participation of multiple researchers in the data interpretation process (Cho & Trent, 2006) – as a means for enhancing clarification and accuracy.

  1. I commend the authors for conducting this important study and anticipate its significant impact on the field. I am confident that the article will garner substantial attention and citation in the future.

Thank you for your positive feedback.

The following references have been added to the revised manuscript:

Dai, D.Y.; Li, X. Behind an Accelerated Scientific Research Career: Dynamic Interplay of Endogenous and Exogenous Forces in Talent Development. Educ. Sci. 202010, 220. https://doi.org/10.3390/educsci10090220

Johansson, S., & Lundqvist, C. (2017). Sexual harassment and abuse in coach–athlete relationships in Sweden. European Journal for Sport and Society14(2), 117–137.‏

Marracho, P., Coelho, E. M. R. T. D. C., Pereira, A., & Nery, M. (2023). Mistreatment behaviours in the Athlete-coach relationship. Retos, 2023(50), 790–7989.‏

McDonald, B., & Kawai, K. (2017, April). Punishing coaching: Bukatsudō and the normalisation of coach violence. Japan Forum, 29(2), 196–217.

Zogg, C. K., Runquist, E. B., Amick, M., Gilmer, G., Milroy, J. J., Wyrick, D. L., ... & Tuakli-Wosornu, Y. A. (2024). Experiences of interpersonal violence in sport and perceived coaching style among college athletes. JAMA Network Open7(1), e2350248–e2350248.‏

Reviewer 3 Report

Comments and Suggestions for Authors

Dear authors, the topic you present is intriguing, and the research object itself is relevant and important for the entire sports community.

The presentation of the abstract is clear and meets the requirements.

In the introductory part, would be nice to see a deeper account of the authors in the analyzed topic. First of all, the topic of violence in sports is analyzed. However, even now, there is no single answer in the scientific literature as to what violence in sport is. If the authors would analyze such phenomena as aggression, bullying, harassment, etc. more deeply in sport and their links with violence in sport, would probably not claim that "... the ratively small number on sports violence..." (line 93). I suggest in this part to analyze these related negative phenomena in sports a bit more deeply and to look for researches of recent years (there are certainly quite a few of them conducted in Portugal, Lithuania). This will also strengthen the introduction of the article, i.e. it will allow to justify the relevance and show the research of the problem, but it will also allow to discuss the research data in a better way.

In the Materials and Methods part, when describing the qualitative study, an explanation is missing of the philosophical attitude the researchers were guided by, to specify the data collection method, explaining its choice, to indicate how the interview guidelines were drawn up (if any), to describe the data processing method, its steps and procedures in more detail.

The results of the quantitative research were presented clearly and widely, but the need to present Table 1 is debatable, since the essence of the research is not individual actions, but certain types of violence. Qualitative research is particularly debatable. What did the authors discover new in their qualitative analysis? The only Theme 3, I think, is worth paying attention to and interesting when discussing violence in sport. Perhaps the authors' lack of competence in thematic analysis prevented them from grasping the deep themes. 

The discussion part lacks deeper insights in the analysis of the obtained results, and this is related to the failure to fully analyze the violence phenomenon in sport.

The conclusion is weak, does not fully reflect the research results and is more of a recommendation nature. I suggest to link it more strongly with the results of the research itself - what is new revealed.

Despite the observed shortcomings, I am convinced that the conducted research is worthy of attention, but it requires a deeper analysis both at the theoretical and empirical levels. 

Author Response

Reviewer 3

  1. In the introductory part, would be nice to see a deeper account of the authors in the analyzed topic. First of all, the topic of violence in sports is analyzed. However, even now, there is no single answer in the scientific literature as to what violence in sport is. If the authors would analyze such phenomena as aggression, bullying, harassment, etc. more deeply in sport and their links with violence in sport, would probably not claim that "... the ratively small number on sports violence..." (line 93). I suggest in this part to analyze these related negative phenomena in sports a bit more deeply and to look for researches of recent years (there are certainly quite a few of them conducted in Portugal, Lithuania). This will also strengthen the introduction of the article, i.e. it will allow to justify the relevance and show the research of the problem, but it will also allow to discuss the research data in a better way.

Thank you for your important input. In the revised manuscript, line 130 now reads: “the relatively small degree of sports violence that has been reported.”

Moreover, you are correct that we do not relate to aggression at all, but to the longitudinal ongoing violence of female and male coaches towards athletes.

Additionally, while we did not find a related study from Lithuania, we have expanded the literature review to included studies from a number of countries, as follows:

A dearth of literature on coach violence in sports can be seen, with similarities and differences between countries and cultures. For example, when presenting athletes and coaches in the UK with a series of vignettes that depicted a coach’s emotionally abusive behavior towards a 14-year-old athlete, Gervis, Rhind, and Luzar (2016) found that perceived competitive levels and athletic performance were negatively associated with perceived abusive coach behaviors. In other words, the higher the perceived athletic performance, the lower the perceived abuse. Additionally, McDonald and Kawai (2017) examined perceptions of punishing coach behaviors among students from ten universities in Japan; these researchers found that participants normalized acts of coach violence and accepted them as necessary forms of discipline – often even interpreting such acts as indicative of coach care and kindness.

In the USA, Zogg, et al. (2024) explored interpersonal violence in sports and perceived coaching styles among more than 4,000 college athletes. The researchers found that almost 10% of the respondents reported having experienced at least one type of interpersonal violence during their college sports careers – even in the weeks immediately leading up to the study. Participants who reported having been exposed to interpersonal violence also reported worse psycho-social outcomes, including increased burnout, an expressed desire to quit their sport, and even decreased emotional well-being – especially among participants who identified as female and with non-heterosexual orientations.

In Portugal, Marracho et al. (2023) asked young athletes about their experience or exposure to undesirable coach behaviors. As victims, no participants reported having experienced mistreatment behaviors in their athlete-coach relationship; yet as observers, many reported having witness various forms of abuse, including verbal and emotional abuse, as well as a lack of attention and support. Finally, in Sweden, Johansson and Lundqvist (2017) explored sexual harassment and abuse, as well as coach–athlete relationships. The participants included more than 450 current and former sports-club athletes, of whom 5.5% reported some form of sexual harassment and abuse by their coach – mainly inappropriate, unpleasant, or offensive physical contact. No differences in coach harassment and abuse were seen between genders, sport performance levels, or individual/team sports. Moreover, 13–39% of the participants reported their having experienced dependency, substantial coach influence over their personal-life, non-instructional physical contact, sexualized comments and jokes, and flirting. Such findings from different countries and regions suggest that coach behaviors and coach-athlete relationships are culturally rooted, and may require major reconsideration of the role of sports in education. between countries could be especially beneficial in future research, to further understand the culturally rooted issue of coach abuse.

  1. In the Materials and Methods part, when describing the qualitative study, an explanation is missing of the philosophical attitude the researchers were guided by, to specify the data collection method, explaining its choice, to indicate how the interview guidelines were drawn up (if any), to describe the data processing method, its steps and procedures in more detail.

Thank you for highlighting this issue. The revised manuscript now reads:

The study was approved by the Institutional Review Board at the authors’ affiliated academic institution [name removed for masked review]. To recruit participants, the questionnaire was sent to all sport unions and federations across Israel via e-mail, asking them to kindly forward it to their past and current athletes who are currently aged 18+ years. The same request was also sent to the Israeli Olympic Committee. In addition, the snowball method was applied via the authors’ personal social-media pages.

The interviews, which were conducted via the Zoom platform, lasted 50–120 minutes and with the participants’ approval were recorded and transcribed. Upon completing the online questionnaire, the interviewees had stated that they would be willing to further discuss the issue of coach violence with the researchers. Our contact details (names, telephone numbers, and e-mail addresses) were included on the form. As with the questionnaire, complete confidentiality was ensured to all interviewees.

Analysis of the qualitative data was conducted by the first two authors, and then reviewed by the additional two authors. The achieved input was analyzed and coded according to anticipated potential categories, based on the theoretical framework of the study – namely the types of violence. In other words, the data from the interviews underwent stages of categorization, including the describing, comparing, and relating of data to existing knowledge, as suggested by Attride-Stirling (2001) and Bazeley (2009). Finally, trustworthiness was established using investigator triangulation (Anney, 2014), i.e., the participation of multiple researchers in the data interpretation process (Cho & Trent, 2006) – as a means for enhancing clarification and accuracy.

  1. The results of the quantitative research were presented clearly and widely, but the need to present Table 1 is debatable, since the essence of the research is not individual actions, but certain types of violence.

We perceive Table 1 as being both relevant and interesting, as it presents the distribution of certain types of violence. Therefore, we would prefer to preserve it in the published article.

  1. Qualitative research is particularly debatable. What did the authors discover new in their qualitative analysis? The only Theme 3, I think, is worth paying attention to and interesting when discussing violence in sport. Perhaps the authors' lack of competence in thematic analysis prevented them from grasping the deep themes. The discussion part lacks deeper insights in the analysis of the obtained results, and this is related to the failure to fully analyze the violence phenomenon in sport. This comment repeats the former, and we were not in agreement with the reviewer. Maybe an example would have helped.The conclusion is weak, does not fully reflect the research results and is more of a recommendation nature. I suggest to link it more strongly with the results of the research itself - what is new revealed.

Following your insightful comment, we have added the following sentences to the revised paragraph:

The novelty of this study’s results is primarily expressed through exploration of the lingering and ongoing impact of coaches’ violence and abuse against athletes. Furthermore, the employed methodology, which integrated both quantitative and qualitative research, enabled the participants to sound their voice and put a so-called face to the numbers. This research approach is relatively rare in studies on violence in sports. As such, the study’s significance lies in its ability to humanize the individuals behind these incidents, particularly children.

The following references have been added to the revised manuscript:

Dai, D.Y.; Li, X. Behind an Accelerated Scientific Research Career: Dynamic Interplay of Endogenous and Exogenous Forces in Talent Development. Educ. Sci. 202010, 220. https://doi.org/10.3390/educsci10090220

Johansson, S., & Lundqvist, C. (2017). Sexual harassment and abuse in coach–athlete relationships in Sweden. European Journal for Sport and Society14(2), 117–137.‏

Marracho, P., Coelho, E. M. R. T. D. C., Pereira, A., & Nery, M. (2023). Mistreatment behaviours in the Athlete-coach relationship. Retos, 2023(50), 790–7989.‏

McDonald, B., & Kawai, K. (2017, April). Punishing coaching: Bukatsudō and the normalisation of coach violence. Japan Forum, 29(2), 196–217.

Zogg, C. K., Runquist, E. B., Amick, M., Gilmer, G., Milroy, J. J., Wyrick, D. L., ... & Tuakli-Wosornu, Y. A. (2024). Experiences of interpersonal violence in sport and perceived coaching style among college athletes. JAMA Network Open7(1), e2350248–e2350248.‏

Round 2

Reviewer 3 Report

Comments and Suggestions for Authors

Dear authors,

Thank you for correcting the theoretical part. Of course, you have not touched on all the authors who have analyzed negative coach-athlete relationships. However, the greatest attention should be paid to the Materials and Methods part. There are still shortcomings in the presentation of qualitative research - an explanation is missing of the philosophical attitude the researchers were guided by, to specify the data collection method, explaining its choice, to describe the data processing method, its steps and procedures in more detail.

You should still work on this part.

Author Response

We would like to thank the editor and reviewer for their second-round comments regarding the manuscript and for enabling us to resubmit a revised version again.

We have implemented the changes as requested within the revised article and have also addressed it below, highlighted in yellow.

Reviewer 3:

An explanation is missing of the philosophical attitude the researchers were guided by, to specify the data collection method, explaining its choice, to describe the data processing method, its steps and procedures in more detail.

We added the following (data collection method and procedure is specified separately in each part – quantitative and qualitative)

The current study uses a Mixed Methods Phenomenological Research (MMPR) approach, specifically an exploratory sequential design. This means it starts with quantitative and continue with qualitative data to make the quantitative assessment more precise (Creswell, 2007). The reason for using MMPR is that phenomenological and quantitative methods can work together well under one main framework. Even though they have different beliefs about reality and knowledge, they share similar values and methods that make combining them possible (Mayoh & Onwuegbuzie, 2015).

Phenomenology, often considered a human science approach, shares scientific nature with deductive approaches, justifying the adoption of an MMPR approach. The flexibility of phenomenological methods, rooted in understanding subjective experiences, complements the strengths of quantitative methods. By incorporating multiple methods within a single study, researchers can enhance validity, minimize bias, and capitalize on the strengths of each method. The integration of quantitative methods can provide orientation, identify relevant phenomena, and test theories developed through phenomenological inquiry. Overall, the philosophical rationale for MMPR is grounded in the idea that combining phenomenological and quantitative methods under a single paradigm can yield a comprehensive and nuanced understanding of research phenomena (Onwuegbuzie, Johnson, & Collins, 2009).

Creswell, J. W. (2007). Qualitative inquiry and research design: Choosing among five approaches.

Thousand Oaks, CA: Sage.

Mayoh, J., & Onwuegbuzie, A. J. (2015). Toward a conceptualization of mixed methods phenomenological research. Journal of mixed methods research9(1), 91-107.‏

Onwuegbuzie, A. J., Johnson, R. B., & Collins, K. M. (2009). Call for mixed analysis: A philosophical framework for combining qualitative and quantitative approaches. International journal of multiple research approaches3(2), 114-139.‏

We also emphasized what appears in yellow in the procedure of the quantitative part:

The study was approved by the Institutional Review Board at the authors’ affiliated academic institution [name removed for masked review]. To recruit participants, the questionnaire was sent to all sport unions and federations across Israel via e-mail, asking them to kindly forward it to their past and current athletes who are currently aged 18+ years. The same request was also sent to the Israeli Olympic Committee. In addition, the snowball method was applied via the authors’ personal social-media pages.

The interviews, which were conducted via the Zoom platform, lasted 50–120 minutes and with the participants’ approval were recorded and transcribed. Upon completing the online questionnaire, the interviewees had stated that they would be willing to further discuss the issue of coach violence with the researchers. Our contact details (names, telephone numbers, and e-mail addresses) were included on the form. As with the questionnaire, complete confidentiality was ensured to all interviewees.

Analysis of the qualitative data was conducted by the first two authors, and then reviewed by the additional two authors. The achieved input was analyzed and coded according to anticipated potential categories, based on the theoretical framework of the study – namely the types of violence. In other words, the data from the interviews underwent stages of categorization, including the describing, comparing, and relating of data to existing knowledge, as suggested by Attride-Stirling (2001) and Bazeley (2009). Finally, trustworthiness was established using investigator triangulation (Anney, 2014), i.e., the participation of multiple researchers in the data interpretation process (Cho & Trent, 2006) – as a means for enhancing clarification and accuracy.

For legal purposes, we chose to only include athletes over the age of 18; had we been exposed to violence against younger athletes, we would have had to report this to the police, according to the Child Protection Law that was enacted in Israel in 1995, and as such, would not have been able to maintain complete confidentiality for the respondents.

Round 3

Reviewer 3 Report

Comments and Suggestions for Authors

Dear authors, I think that your work on the correction of the article really contributed to its quality. Good luck with your further research.